# Determination of the Degree of Penetration of Glass Ionomer Cements in the Healthy and Decayed Dentine of Permanent Molars

**DOI:** 10.3390/ma18173984

**Published:** 2025-08-25

**Authors:** Pilar Valverde-Rubio, Pilar Cereceda-Villaescusa, Inmaculada Cabello, Andrea Poza-Pascual, Clara Serna-Muñoz, Antonio José Ortiz-Ruiz

**Affiliations:** 1Department of Integrated Pediatric Dentistry, Faculty of Medicine, Universidad Católica San Antonio de Murcia, 30107 Guadalupe, Murcia, Spain; mpvalverde@ucam.edu; 2Department of Integrated Pediatric Dentistry, Faculty of Medicine, University of Murcia, 30008 Murcia, Spain; pilar.cerecedav@um.es (P.C.-V.); ajortiz@um.es (A.J.O.-R.); 3Department of Integrated Pediatric Dentistry, Universidad de Granada, 18011 Granada, Spain; icabello@ugr.es; 4Department of Integrated Pediatric Dentistry, Universidad del País Vasco, 48940 Leioa, Bizkaia, Spain; andrea.poza@ehu.eus

**Keywords:** glass ionomer, dentin, minimal intervention

## Abstract

This study aimed to evaluate the penetration and bonding performance of three restorative materials—high-viscosity glass ionomer cement (Riva Self Cure HV), resin-modified glass ionomer cement (Riva Light Cure) and a bioactive resin (Activa BioActive Restorative™)—in the healthy and carious dentine of permanent molars. Forty extracted human molars with sound or decayed dentine were restored following standardised protocols and subsequently divided into slices. So, twenty-four samples were used for each group (sound and carious dentine) for interface analysis using confocal laser scanning microscopy, field emission scanning electron microscopy and energy dispersive X-ray spectroscopy, and another eight simples were used for each group (sound and carious dentine) for Vickers microhardness testing. Results showed that both glass ionomer cements achieved consistent chemical bonding in healthy dentine and demonstrated better interfacial adaptation compared to carious dentine, where partially demineralised areas showed weaker bonding. The bioactive resin exhibited good adhesion in sound dentine due to the adhesive system but showed poorer interaction in decayed dentine with signs of interfacial separation. Elemental analysis revealed similar compositions among materials, with no significant differences in material concentrations among the ionomers, while there were significant differences with the other materials. On the other hand, some variations were observed in the sulphur, fluoride and strontium content depending on dentine condition. Microhardness values were higher in healthy dentine than in carious dentine for all materials (*p* < 0.001), except the high-viscosity glass ionomer, which maintained stable hardness in both substrates (36.33 ± 6.23 VHN vs. 34.56 ± 4.31 VHN; *p* = 0.605). These findings highlight the relevance of material selection and dentine condition in minimally invasive restorative dentistry.

## 1. Introduction

Adhesion between restorative materials and tooth tissue is a complex process, influenced by various factors such as the morphological and structural characteristics of the substrate, the intrinsic properties of the materials used and the nature of the interaction between the two [1,2,3,4]. In this context, dentine represents a real challenge due to its highly complex composition and structure, which varies according to its anatomical location and state of mineralisation [1,2].

From a structural point of view, dentine is a three-dimensional matrix composed mainly of type I collagen fibres, arranged in parallel and connected to each other by non-collagenous proteins, in which small hydroxyapatite crystals are embedded [5]. In addition, it contains a wide range of ions, including calcium, phosphorus, fluoride, zinc and strontium [6], which play an important role in remineralisation processes. In situations of partial demineralisation, still-intact collagen fibrils can be observed, accompanied by residual minerals, which opens up the possibility of therapeutic intervention using remineralizing materials [7].

With the advance of minimally invasive dentistry (MID), a new clinical paradigm has emerged that seeks to preserve as much healthy dental tissue as possible, halt the progression of caries and maintain pulp vitality [8,9,10,11]. This approach drives the search for and development of new materials capable of biocompatible and effective integration with dentine, especially partially demineralised dentine. Ideal candidates include adhesive and bioactive materials with remineralising capacity [12].

Among these, glass ionomer cements (GICs) have established themselves as a valuable option due to their distinctive properties: fluoride release, chemical bonding to enamel and dentine, biological compatibility and a tooth-like coefficient of thermal expansion. However, their clinical use presents challenges, especially because of their sensitivity to moisture during setting and their low resistance to wear and fracture [13,14].

To overcome these limitations, variations have been developed such as high-viscosity glass ionomer cements (HV GIC), which incorporate significant changes in their composition: a higher powder–liquid ratio, higher molecular weight polymers and reactive fillers such as fluoro-alumino-silicate (FAS), which improve both their hardness and adhesion [3,15]. Resin-modified GICs (RM GICs) integrate hydrophilic resin monomers and photoinitiation systems, which enhance their mechanical properties and improve their moisture stability [13,15,16].

The ability of these materials to release ions—mainly fluoride, calcium and aluminium—gives them outstanding anti-cariogenic activity. This bioactive behaviour, which allows for the exchange of ions with the oral environment, positions them as key tools within the MID philosophy [13,15,17].

According to the manufacturer, Riva glass ionomer cements (SDI) are composed of multiple ultrafine particles of varying sizes, which ensure restorations of good strength and aesthetics, as well as easy handling (no etching or adhesive is required, as they provide chemical bonding). In addition, they contain fluoride and strontium ions that improve the biomineralisation of tooth tissue.

Against this backdrop, innovative materials are emerging that seek to combine the best of GICs and RCs while minimizing their disadvantages. One of the most representative examples is Activa BioActive Restorative™ (Pulpdent Corp., Watertown, MA, USA), a material that does not contain BPA or its derivatives, and whose composition includes a bioactive resin matrix reinforced with silanised fillers and dual activation (chemical and photonic) [15,18,19]. Its ability to release and recharge ions such as calcium, phosphate and fluoride, together with its moisture tolerance and mechanical strength, make it a promising material for restorations in the context of minimal intervention [19]. However, there are discrepancies between the manufacturer’s claims and the scientific literature, which states that the bioactivity of a materials is its ability to induce a biological response, which, to date, has not been demonstrated; furthermore, the adhesive system used to improve the mechanical properties of Activa BioActive Restorative™, may act as a barrier between the material and the tooth tissue, possibly limiting or preventing the bioactive effect by preventing direct contact between the two [15,20,21].

Since the microstructure and composition of healthy dentine differs significantly from carious dentine, adhesion with these materials also varies [4,22,23]. Furthermore, the type of material influences the bonding mechanism: while resins require an adhesive system to achieve a micromechanical bond, GICs do so mainly through a direct chemical interaction with the tissues [2,24].

In this investigation, it is therefore the null hypothesis that the mechanical behaviour of the carious dentine—resin-modified glass ionomer and bioactive resin interfaces is no better than when the glass ionomer is a conventional high-viscosity glass ionomer.

The main objective of this work is to analyse, from a morphological, chemical and mechanical perspective, the bonding interfaces generated between healthy and carious dentine and three different materials: a high-viscosity glass ionomer (Riva Self Cure HV), a resin-modified glass ionomer (Riva Light Cure) and a bioactive resin (Activa BioActive Restorative™). Techniques such as confocal laser scanning microscopy, field emission scanning electron microscopy, energy dispersive X-ray spectroscopy and Vickers surface microhardness measurement will be used.

## 2. Materials and Methods

### 2.1. Selection and Initial Storage of Teeth

A total of 40 human permanent molars extracted for periodontal reasons at the Dental Clinic of the University of Murcia were used. Of these, 20 had healthy dentine without carious lesions and 20 showed occlusal–proximal caries classified as code 5 according to the International Caries Detection and Assessment System (ICDAS), characterised by a cavity with dentine visible up to half of the tooth surface without pulp involvement. All patients gave informed consent for the use of their teeth in this study. After extraction, the teeth were cleaned, rinsed with distilled water and immersed in 0.1% thymol solution for 24 h [24], then stored at 4 °C in distilled water renewed daily, and used within 6 months after extraction. The protocol was approved by the Research Ethics Committee of the University of Murcia (ID: CBE 632/2024) (Appendix A).

### 2.2. Experimental Groups

The teeth were randomly assigned to the different experimental groups using a table of random numbers generated with the Excel programme (=ALEATORIO.ENTRE) (Table 1).

### 2.3. Sample Preparation

Cavity preparation:

In teeth with healthy dentine, occlusal–proximal cavities of approximately 4 × 4 × 4 mm, located above the amelocementary line, were made using a tapered bur 6830L.314. 012 (Komet Dental, Gebr. Brasseler GmbH & Co. KG, Lemgo, Germany) in a Synea Vision TK-94 turbine at 360,000 rpm (W&H Dentalwerk GmbH; Salzburg, Austria). In teeth with cavitated lesions, enamel and carious dentine were removed from the margins and irreversibly demineralised dentine at the bottom of the cavity was removed with an LM excavating spoon (Parafinen, Finland).

Application of restorative materials:

For restorations with Riva Light Cure^®^ and Riva Self Cure HV^®^ (SDI), the surface was washed with distilled water spray, dried with compressed air without desiccation and Riva Conditioner (polyacrylic acid) was applied for 10 s. After rinsing and maintaining light moisture, the capsule was activated and mechanically mixed for 10 s at 4550 rpm. For subsequent visualisation, Rhodamine B 0.05% by weight (RhB; Sigma-Aldrich Chemie Gmbh, Riedstr, Germany) [23] was added to the mixture manually with a spatula. Riva Light Cure^®^ was applied in maximum 2 mm increments and light cured 20 s per layer with the Demi™ Ultra LED lamp (Kerr, CA, USA), finishing with the application and light curing of Riva Coat^®^.

Riva Self Cure HV^®^ was applied in a single increment and after curing, Riva Coat^®^ was applied and cured.

For Activa BioActive Restorative™ (Pulpdent), the surface was treated with 37% orthophosphoric acid (Dentaflux, Madrid, Spain) for 15 s after washing and light drying, followed by the application of Prime&Bond Active™ adhesive (Dentsply Sirona; Charlotte, NC, USA, EE. UU.) by rubbing for 20 s and light curing for 20 s. The material was dispensed with an automix syringe, and the Rhodamine B [23] was added and applied in increments of up to 4 mm, light curing each for 20 s.

For the control group, Grandioso^®^ composite resin (Voco) was used, the procedure for this being the same as for Activa BioActive Restorative^TM^, with acid etching, Prime&Bond Active™ adhesive and application of the material with Rhodamine B in 2 mm increments, followed by light curing.

### 2.4. Application of Fluorescein

Fluorescein was used in two ways:(a)In groups IAa, IIAa, IIIAa, IVAa, IBa, IIBa, IIIBa and IVBa (Table 1), after restoration, the middle and lower third of the molar roots were sectioned down to the furcation with a W-H turbine (Bürmoos, Austria) and diamond-tipped truncated cone bur, removing pulp tissue and sealing with nail polish [16], leaving access only to the pulp chamber, which was filled with 1% aqueous fluorescein/ethanol using a syringe (Sigma-Aldrich Chemie Gmbh, Riedstr, Germany). Subsequently, the samples were immersed in fluorescein for 3 h [16,23].(b)In groups IIIAb, IIIBb, IVAb and IVBb (Table 1), the fluorescein was mixed directly with the adhesive before application, without injection or subsequent dipping.

All samples were stored for 24 h in distilled water at room temperature.

### 2.5. Cutting and Final Preparation of the Samples

After 24 h of soaking in distilled water, the prepared tooth crowns were placed in plastic containers where they were immersed in Aka-Resin self-curing epoxy resin (AKASEL A/S; Roskilde, Denmark) with a 100/12 weight ratio of Aka-resin liquid epoxy and Aka-cure slow (catalyst), respectively. The containers were vacuumed for 2 h with a vacuum machine N 86 Laboratory (Village-Neus, France) and then placed at 37 °C for 24 h in an Incubat Safety Thermostar (JP Selecta, Abrera, Barcelona, Spain).

The teeth were removed from the plastic containers and cut longitudinally in the vestibulo-palatal/lingual direction with a precision saw (Precision Saw Isomet 1000^TM^Buehler^®^, Lake Bluff, IL, USA. Subsequently, water-cooled, they were polished on both sides on the Saphir 250 M1 polishing machine (ATM GmbH, Mammelzen, Germany) using SiC (carbon–silicon) abrasive discs of 1200 to 4000 grits and velvet disc. For each tooth, 2 slices of approximately 1 mm were obtained and ultrasonically cleaned for 5 min (BioSonic UC50DB, Coltene; Ascot Parkway, Cuyahoga Falls, OH, USA) immersed in distilled water. The samples were stored in a humidity chamber [23].

### 2.6. Confocal Laser Scanning Microscopy (CLSM)

For CLSM (confocal laser scanning microscopy), FESEM (field emission scanning electron microscopy) and EDX (energy dispersive X-ray), the same samples were used: 2 teeth per group, two slides per tooth. Each slide was placed on a 26 × 76 mm glass slide with a drop of 80% glycerol and 20% water added and protected with a 24 × 24 mm glass coverslip for observation. The analysis of the bonding interfaces of the material to the dentine was performed with a Stellaris 8 (Leica, Heidelberg, Germany) spectral scanning confocal microscope equipped with a white laser, dry lenses (10×) and immersion lenses (63× oil and glycerol, 86× water and 100× oil). The samples were first observed with the 10×/0.4 dry objective and then with the 63×/1.30 objective with glycerol as the immersion medium. The evaluation of the diffusion of the materials was performed with a Rhodamine excitation laser, which was excited using green light (540 nm) and emitted in red (590 nm), and the fluorochrome fluorescein, which was activated with blue light (495 nm) and emitted in green–yellow (520 nm). Each interphase was fully visualised and micrographs, representing the most common morphological features observed, were taken using the Leica “Las X” software version 3.7 (Heidelberg, Germany).

### 2.7. Field Emission Scanning Electron Microscopy (FESEM)

The same samples that were observed in CLSM were used for study in FESEM. For this, they were ultrasonically cleaned and immersed in 100% ethanol for 15 min to remove glycerol residues. They were then left to dry at room temperature on blotting paper for 24 h and attached by carbon tape to a platen. They were platinum coated (5 nm layer) in the Leica EM ACE600 metallisation chamber (Leica, Heidelberg, Germany). After this process, they were kept in a desiccator until they were viewed under the FESEM Zeiss Crossbeam 350 microscope (Carl Zeiss Microscopy GmbH, Oberkochen, Germany).

The parameters used in the FESEM visualisation were a 5 mm working distance and a 10 KV accelerating voltage. Zeiss SmartSEM 6.07 (2020) software was used to obtain the most representative images.

### 2.8. Energy Dispersive X-Ray (EDX) Analysis and Mapping

To determine the elemental composition, EDX analysis was performed at a 10 kV accelerating voltage using the Oxford Instrument EDX detector (Abingdon, Oxfordshire, UK) and AZtec software version 5.0.7577.2 (2020). Data were expressed as a percentage by weight (*w*/*w*). The areas where EDX analysis was performed were as follows: (a) one area in the material immediately adjacent to the interface; (b) three areas in the dentine immediately adjacent to the interface; and (c) one area in the adhesive, in the samples where it was used. In addition, elemental mapping was performed on a single sample per group to determine the distribution of the elements in the dentine, material and adhesive.

### 2.9. Vickers Microhardness

Two teeth per group were used—1 slide per tooth. The slices were fixed by means of an adhesive tape on the surface of the plate, which was placed on the Vickers KMHV-1000Z benchtop microhardness tester (Kansert S.L., Orkoien, Navarra, Spain). First, it was focused at 10× to locate the interface and then at 40× to select the exact areas where the indentations were to be made; these were carried out in three areas: material, dentine and at the material–dentine interface. In each area, three measurements were taken, separated by three times the size of the indentation mark, and the Vickers hardness (VHN) values were averaged. The indentations were carried out with a load of 300 g for 10 s. Micrographs of the samples subjected to the microhardness test were taken using a Stemi 305 Axiocam 208 colour microscope (Carl Zeiss Microscopy GmbH, Germany).

### 2.10. Statistical Analysis

The variables analysed in this study included the following: the weight percentage concentration of the chemical elements present in the adhesive; the weight percentage concentration of the elements in each restorative material; and the elemental composition of sound and carious dentine adjacent to the adhesive interface. Additionally, microhardness was assessed at the material–dentine interface (both sound and carious), within the restorative materials and in the sound and carious dentine themselves.

Statistical analyses were performed using Jamovi version 2.3. Descriptive statistics were computed for all study variables. Shapiro–Wilk tests were used to assess normality, and Levene’s test was applied to evaluate the homogeneity of variances. To assess differences in material composition, a one-way ANOVA was conducted followed by Tukey’s post hoc test when the assumptions of normality and homoscedasticity were met (elements: O, C, Al). For elements that did not meet these assumptions (Si, Na, Ca, Sr, F, Ba), the Kruskal–Wallis test followed by the Dwass–Steel–Critchlow–Fligner test was used.

To examine differences in dentine composition near the adhesive interface, the same approach was applied. A one-way ANOVA with Tukey’s post hoc test was used when assumptions were satisfied (sound dentine: O, Ca, Na, Mg; carious dentine: O, P, N, Na, F), and the Kruskal–Wallis test with Dwass–Steel–Critchlow–Fligner post hoc analysis was used when assumptions were violated (sound dentine: C, N, F, P; carious dentine: Ca, C, Sr, Mg, S, Si).

As microhardness values across the experimental groups met the assumptions of normality and the homogeneity of variance, a one-way ANOVA followed by Tukey’s test was used to identify pairwise differences. Differences in the microhardness between sound and carious dentine and the material–sound dentine and carious dentine interfaces were evaluated using an independent samples *t*-test.

A *p*-value of <0.05 was considered statistically significant. 

## 3. Results

### 3.1. Evaluation of the Morphology of the Healthy Dentine Interface with Riva Light Cure, Riva Self Cure HV, Activa BioActive Restorative™ and GrandioSO^®^ Restorative Materials

CLSM and FESEM images are shown for the healthy (Figure 1) and carious (Figure 2) dentine groups. 

#### 3.1.1. Group IAa (Riva Light Cure—Fluorescein Injected Through the Pulp Chamber)

CLSM revealed the formation of a continuous interface between the dentine and Riva Light Cure, evidenced by Rhodamine staining (asterisk) and the presence of tags in the dentinal tubules. Green areas of demineralisation not infiltrated by the material were observed (white arrow). In FESEM, a cohesive fracture of the RM GIC, the hybrid zone with dentine, irregular crystalline particles of FAS and pores of various sizes were distinguished.

#### 3.1.2. Group IIAa (Riva Self Cure HV—Fluorescein Injected Through the Pulp Chamber)

CLSM showed a cohesive fracture of the ionomer (f) and areas of demineralisation without infiltration (arrow). However, in FESEM, an intimate bonding of the material with dentine was evident (asterisks). Morphologically, large filler particles and pores of different diameters were observed.

#### 3.1.3. Group IIIAa and IIIAb (Activa BioActive Restorative™—Fluorescein per Pulp Chamber and in the Adhesive)

The images showed good adaptation between the dentine, adhesive and material (white marker), with presence of a rich matrix and small particles, as well as smaller pores. In IIIAb, numerous resin tags and intertubular connections were observed.

#### 3.1.4. Group IVAa and IVAb (GrandioSO^®^—Fluorescein in the Pulp Chamber and in the Adhesive)

For group IVAa, in CLSM, good interfacial adhesion and the presence of tags in tubules were observed; in FESEM, excellent bonding between the dentine, adhesive and material was evidenced, with a cohesive fracture of dentine (f) and some punctual mismatches (black marker). GrandioSO^®^ showed a homogeneous matrix with microparticles, some isolated pores and a continuous hybrid layer (HL) with obvious tags when fluorescein was incorporated into the adhesive (IVAb), but with areas of non-adaptation (black marker).

### 3.2. Evaluation of the Morphology of the Carious Dentine Interface with Riva Light Cure, Riva Self Cure HV, Activa BioActive Restorative™ and GrandioSO^®^ Restorative Materials

#### 3.2.1. Group IBa (Riva Light Cure—Fluorescein Injected Through the Pulp Chamber)

In CLSM, a cohesive fracture of carious dentine with retention of the bonded material (asterisk) and the presence of ionomer tags in dentinal tubules was observed. In FESEM, healthy and carious dentine zones were differentiated.

#### 3.2.2. Group IIBa (Riva Self Cure HV—Fluorescein Injected via the Pulp Chamber)

Adhesion was dependent on the degree of demineralisation of the carious dentine. In partially destructured areas, bonding of the material to the substrate was evident (white marker), whereas in areas with structural loss, no bonding was observed. In FESEM, in the bonded regions, cohesive fractures of the GIC were identified (asterisk).

#### 3.2.3. Group IIIBa and IIIBb (Activa BioActive Restorative™—Fluorescein per Pulp Chamber and in the Adhesive)

Interfacial separation between the carious dentine and Activa BioActive was identified, due to the adhesive and, above all, a cohesive fracture of the carious dentine. In IIIBb, the fluorochrome allowed the visualisation of an adhesive layer and little tag formation. FESEM analysis confirmed the separation as well as the transition between healthy and carious dentine.

#### 3.2.4. Group IVBa and IVBb (Activa BioActive Restorative™—Fluorescein per Pulp Chamber and in the Adhesive)

An HL of variable thickness was observed. Adhesive-composite bonding was good (white marker), while bonding with carious dentine showed adhesive fractures (black marker) and the presence of fluorochrome in the separation areas (arrow). In FESEM, a dentine fracture was evidenced as the main cause of separation, although areas with good hybridisation were also identified (asterisk).

### 3.3. Evaluation of the Semi-Quantitative Composition of the Interface Between Healthy or Carious Dentine and the Restorative Materials Riva Light Cure, Riva Self Cure HV, Activa BioActive Restorative™ and GrandioSO^®^ by Energy Dispersive X-Ray Spectroscopy Mapping

EDX analysis revealed the common ions mostly present in all materials were oxygen, carbon, silicon and aluminium, with strontium only in Riva Light Cure and Riva Self Cure HV, barium in Activa BioActive Restorative™ and GrandioSO^®^ and fluorine absent in GrandioSO^®^. However, there were significant differences in the elemental concentration of the restorative materials, except between Riva Light Cure and Riva Self Cure HV, where no significant difference was found. In a more specific analysis, there was a statistically higher concentration of carbon in Activa BioActive Restorative™ (*p* < 0.001) and aluminium, sodium and fluoride in Riva Light Cure and Riva Self Cure HV (*p* < 0.001). At the interface with healthy dentine, significant differences in nitrogen, magnesium, sodium and fluoride were observed, and in carious dentine, only fluoride and strontium varied significantly between the groups. Elemental mapping confirmed these results, showing that the GICs presented a clear distribution of elements such as aluminium, silicon, strontium and fluorine in the material and calcium and phosphorus in the dentine, while Activa BioActive Restorative™ and GrandioSO^®^ were dominated by barium, a high concentration of carbon in the adhesive and no fluorine in GrandioSO^®^, although it was detected in the areas of carious dentine. All results can be seen in the tables (Table 2, Table 3, Table 4 and Table 5).

### 3.4. Vickers Microhardness Analysis

Activa BioActive Restorative™ showed the lowest microhardness value, while GrandioSO^®^ showed the highest, with significant differences also compared to Riva Light Cure and Riva Self Cure HV, which differed from each other. Sound dentine had a higher surface hardness than carious dentine (56.93 ± 7.24 VHN vs. 48.05 ± 11.71 VHN; *p* = 0.005). In the sound dentine groups, Riva Light Cure and GrandioSO^®^ showed the highest microhardness values at the interface, higher than Riva Self Cure HV and Activa BioActive Restorative™. However, in the carious dentine groups, Riva Self Cure HV stood out with better microhardness at the interface compared to the other materials. In general, the microhardness at the interface was lower in the carious dentine groups than in the sound dentine groups for all materials, except for Riva Self Cure HV, which maintained similar values in both dentine types (Table 6).

## 4. Discussion

This research compared four types of restorative materials, applied to both healthy and decayed dentine, each with distinct properties that may offer specific advantages depending on the condition of the dentine substrate. The methodology was supported by previous studies that evaluated the interface of materials such as GIC and RM GIC, such as the work of Sidhu and Watson (1998) [16], which, however, only addressed healthy dentine and not including carious dentine, which was the subject of study in research such as that of Toledano et al. (2017) [23]. However, the latter used a staining system for caries removal that differs from our approach. While both studies were based solely on CLSM, our research incorporated additional techniques such as FESEM, EDX and microhardness testing, previously employed in other works [3,7,17,19,25,26,27,28,29,30].

The use of fluorochromes does not have a standardised protocol. In this study, we used two different systems in order to observe whether the use of one or another system influenced the interpretation of the results: the introduction of fluorescein through the pulp chamber for 3 h [16,17], and the combination with Rhodamine B mixed with each restorative material [23,26], which facilitated the visualisation of the tooth structure and the restorative material, reducing possible misinterpretations, and is therefore widely used in CLSM [31]. In groups III and IV, a subgroup b was also added where fluorescein was mixed only with the adhesive [31], which significantly improved the visualisation of the interphase morphology, allowing the tags and intertubular connections to be observed more clearly (Figure 1, CLSM group IIIAb). Since there may be interpretative alterations, it is highly recommended to continue with comparative studies between the different fluorochrome incorporation systems to establish a standardized methodology that will lead to homogeneous interpretations of the results.

Regarding the microhardness study, the literature showed variability in the number of indentations, the type of specimen, load applied and time employed [7,25,28,32]. However, Fuentes et al. (2003) [25] indicated that there were no statistically significant differences attributable to the type of load.

In the images obtained from groups IA and IIA on healthy dentine (Figure 1), it was observed that, despite cohesive fractures in the material, part of the material remained adhered to the substrate. This situation coincides with previous reports indicating that the interfacial bond strength exceeds the cohesive strength of the material itself [33], or that the observed fractures could be artefacts originated during the preparation of the samples for SEM, without affecting the fit and adaptation between the material and healthy dentine [2,3].

In carious dentine, groups IB and IIB (Figure 2) also showed hybridisation of the material to the unstructured substrate despite partial fractures. However, in contrast to our results, Toledano et al. (2017) [23] found that GIC showed good hybridisation in sound dentine but very poor hybridisation in carious dentine, whereas RM GIC showed better performance in both substrates. The differences between the healthy and carious dentine interface in our groups I and II can be attributed to the greater depth of the cavities in the carious samples, as the adhesion in surface dentine is superior due to the lower tubular density and higher mineral content available for chemical interaction [2]. In addition, demineralisation and lower permeability of the tubules in carious dentine negatively affect adhesion [22].

Our results differ from those of Ebaya et al. (2019) [19], who found no significant difference in the marginal adaptation between GIC, RM GIC and Activa BioActive Restorative™ by SEM. However, we concur with studies reporting the similar marginal adaptation of Activa BioActive Restorative™ to RCs when used with adhesive [20,29]. The inclusion of adhesive, as in our work, responds to clinical evidence that showed that the exclusive use of acid etching, without adhesive, generated high rates of restoration loss and secondary lesions, due to the removal of ions necessary for chemical bonding of the material [18].

In our research, Activa BioActive Restorative™ with adhesive showed good hybridisation in healthy dentine, similar to GrandioSO^®^ (Figure 1, groups IIIA and IVA, respectively), but both materials performed worse on carious dentine (Figure 2, groups IIIB and IVB, respectively), where adhesion was weaker due to demineralisation, collagen de-structuring and lower tubular permeability, in addition to the tendency for cohesive fractures favoured by polymerisation shrinkage [4,9,22].

Although the manufacturer of Activa BioActive Restorative™ classifies it as a bioactive resin, this study agrees with Francois et al. (2020) [15] that it should be considered within the group of RM GICs. The term bioactive has been used indiscriminately, confusing real biological activity with remineralizing properties. A bioactive material should induce a proven biological response, something that has not been demonstrated in Activa BioActive Restorative™ either in vitro or in clinical practice [15,21]. Furthermore, the use of adhesive could limit this hypothetical bioactivity [20].

Microhardness analysis showed statistically significant differences between the four materials (Table 6), although this did not match the lower values reported by Hershkovitz et al. (2020) [28] at higher loadings. However, similar results were observed in studies with lower loadings for Activa BioActive Restorative™ and Riva Light Cure HV [30,32]. The higher microhardness of Riva Self Cure HV compared to Riva Light Cure could be due to the larger FAS particle size, which confers higher wear resistance [28].

Microhardness was significantly higher in healthy versus carious dentine, consistent with structural and compositional differences resulting from demineralisation, higher water absorption and the lower elastic modulus of carious dentine, factors that predispose for cohesive fractures [7,9,22]. This was reflected in our study with a higher presence of fractures in the groups that restored carious dentine (IIIB and IVB) compared to healthy dentine (IIIA and IVA).

Regarding interfacial analysis, measurements varied according to the material and dentine type, with a significant decrease in microhardness at the interface with carious dentine, except for Riva Self Cure HV (Table 6). This exceptional behaviour could be attributed to the higher chemical bonding, which would compensate for the lower mineralisation of the carious substrate [3]. In contrast, materials that rely mostly on micromechanical bonding, such as Activa BioActive Restorative™ and GrandioSO^®^, showed poorer performance in carious dentine, where higher humidity and collagen destructuring hinder tag formation [4,22]. In addition, other authors reported that conventional GICs maintain similar adhesive strengths in healthy and carious dentine [11].

In carious dentine, the amount of remaining substrate influenced adaptation; in the adhesive groups (IIIB and IVB), adaptation was generally poor, except when the amount of remaining dentine was low, favouring a higher bond strength to unstructured tissue and a tighter peripheral seal (Figure 2, CLSM IVBa). This supports the recommendation to choose the restorative material considering the location, extent and activity of the lesion, as well as the caries risk and the conditions of the oral environment [9].

In elemental analysis, the majority of the organic ions in the GICs were similar, with percentage differences comparable to previous work [27,28]. GrandioSO^®^, compared to Spectrum, showed a similar composition except for the presence of zirconium instead of barium, with a similar concentration of silicon, but differences in oxygen and carbon [28].

Variations in the concentrations of some ions were observed, without significant changes in the overall composition. The absence of carbon differences between Riva Light Cure and Riva Self Cure HV was striking, possibly due to the compensation between the carboxyl groups and resin. In GrandioSO^®^, all carbon would come from the resin, whereas in Activa BioActive Restorative™, carbon was detected from carboxyl groups and the resin matrix (Table 3).

EDX analysis in healthy dentine showed mainly oxygen, calcium, carbon, phosphorus and nitrogen, together with trace elements such as magnesium, sodium and fluoride, while silicon, strontium and sulphur, possibly derived from diet, dental products or experimental procedures, were detected in carious dentine [6]. Fluoride was present in healthy dentine for all fluoride-containing materials except GrandioSO^®^ and increased in carious dentine, even in samples restored with the latter, suggesting an active absorption of fluoride from the medium, favouring remineralisation (Table 4 and Table 5).

Taking into account the results obtained in this investigation at morphological, chemical and mechanical levels, it could be considered that clinically, in healthy dentine, none of the three materials stood out above the rest in all the parameters analysed. The bioactive resin together with the adhesive showed very good adaptation at the interface with healthy dentine, but it was the one that obtained the lowest microhardness values; however, the glass ionomers showed good bonding and better results in microhardness, especially the RM GIC. On the other hand, in carious dentine, the ionomers were more stable with respect to their interfacial hybridisation, and HV GIC obtained the highest microhardness value in contrast to the bioactive resin with the adhesive, which showed the lowest microhardness record and very poor bonding. All materials obtained higher fluoride concentrations in carious dentine than in healthy dentine.

Considering the MID philosophy, which urges the realisation of our restorations on demineralised dentine substrate, HV GIC could be a material with good behaviour as it presents the best combination of adhesion and microhardness, as well as possessing ions that favour the remineralisation process.

This work has certain limitations that should be considered when interpreting the results. Firstly, the method for fluorochrome application is not standardised, which may introduce variability in the visualisation of the interfaces. In addition, variability in the depth and extent of the carious lesions in the samples could influence the adhesion and microhardness obtained, limiting the generalisability of the findings. On the other hand, the incorporation of adhesive in some groups could have affected the potential biological activity of materials classified as bioactive, preventing the complete evaluation of this property. Finally, because the sample size was not very large, the analyses were performed in vitro, so the results could differ in real clinical conditions where additional environmental and biological factors influence the behaviour of restorative materials; further research is needed to establish more accurate conclusions.

## 5. Conclusions

In this study, the morphological, chemical and mechanical interactions of different restorative materials on healthy and decayed dentine were evaluated.

It was observed that both high-viscosity glass ionomer cement and resin-modified glass ionomer cement generated similar interfaces in both types of dentine, although with better hybridisation in healthy tissue. The bioactive resin, on the other hand, showed good adhesion in healthy dentine due to the use of an adhesive system, but poor interaction in carious dentine, with a general lack of bonding.

At the chemical level, the three materials shared a similar elemental composition, with only some variations in the percentages of elements such as nitrogen, magnesium, sodium, fluorine, strontium and sulphur, depending on the type of dentine.

In the microhardness analysis, the resin-modified glass ionomer cement showed the highest values in the healthy dentine, whereas in the carious dentine, it was the high-viscosity glass ionomer cement that showed the highest hardness. The bioactive resin remained with intermediate values in both conditions, being comparable to the modified ionomer cement in the carious dentine, but lower in the sound dentine.

Finally, the null hypothesis is accepted, as the mechanical behaviour of the carious dentin–high-viscosity glass ionomer cement interface was better than that of the resin-modified glass ionomer cement and bioactive resin.

## Figures and Tables

**Figure 1 materials-18-03984-f001:**
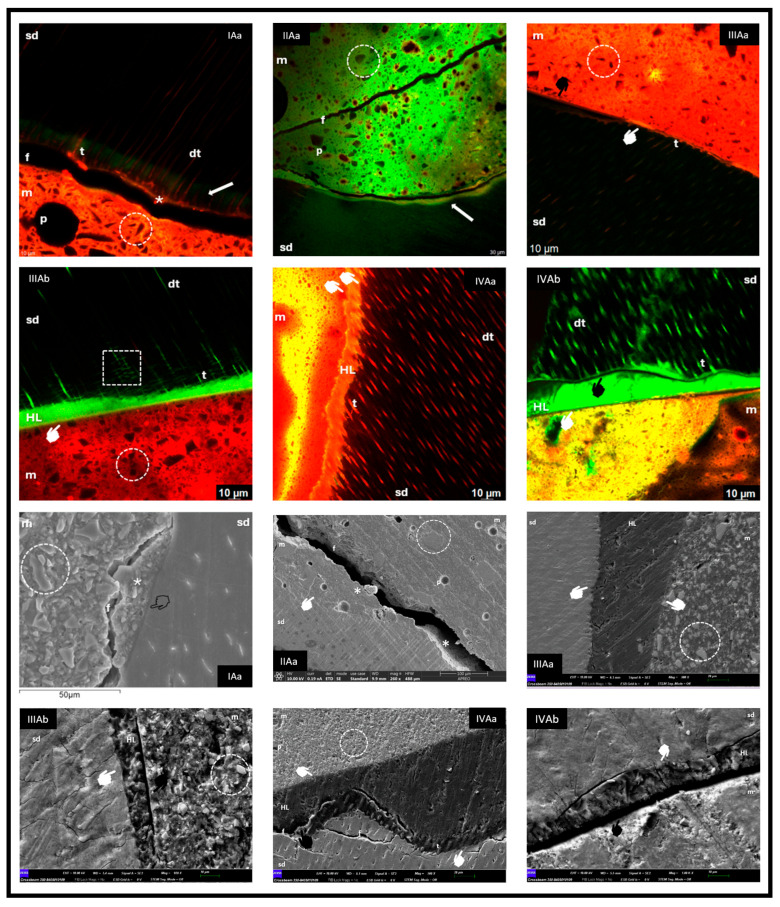
sd: sound dentine; m: material; p: pore; dt: dentine tubule; t: tag; arrow: fluorescein accumulation; circle: material morphology; *: dentine-bonded material; f: fracture; white marker: adhesion at interface; black marker: no adhesion at interface; transparent marker: hybrid zone; HL: hybrid layer; square: intertubular connections.

**Figure 2 materials-18-03984-f002:**
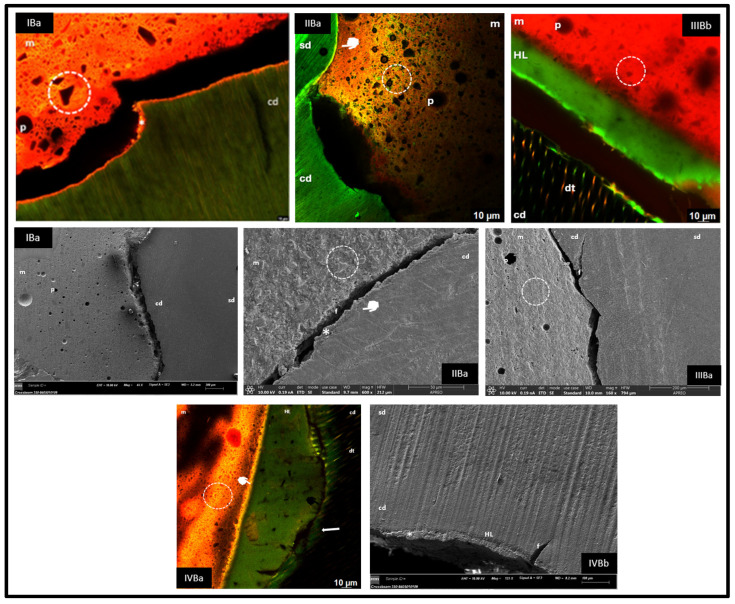
sd: sound dentine; cd: carious dentine; m: material; p: pore; circle: material morphology; *: dentine-bonded material; f: fracture; white marker: bond at interface; black marker: no bond at interface; HL: hybrid layer; arrow: fluorescein accumulation.

**Table 1 materials-18-03984-t001:** Experimental groups.

Group	Description	Fluorochrome	Technique/Sample
(IAa)	Healthy Dentine + Riva Light Cure	Rhodamine B + fluorescein in pulp chamber	CLSM, FESEM, EDX: 2 teeth—2 slices/tooth Microhardness: 2 teeth—1 slice/tooth
(IIAa)	Healthy Dentine + Riva Self Cure HV	Rhodamine B + fluorescein in pulp chamber	CLSM, FESEM, EDX: 2 teeth—2 slices/tooth Microhardness: 2 teeth—1 slice/tooth
(IIIAa)	Healthy Dentine + Activa BioActive Restorative™	Rhodamine B + fluorescein in pulp chamber	CLSM, FESEM, EDX: 2 teeth—2 slices/tooth Microhardness: 2 teeth—1 slice/tooth
(IVAa)	Healthy Dentine + GrandioSO^®^	Rhodamine B + fluorescein in pulp chamber	CLSM, FESEM, EDX: 2 teeth—2 slices/tooth Microhardness: 2 teeth—1 slice/tooth
(IIIAb)	Healthy Dentine + Activa BioActive Restorative ™	Rhodamine B + fluorescein in the adhesive	CLSM, FESEM, EDX: 2 teeth—2 slices/tooth
(IVAb)	Healthy Dentine + GrandioSO^®^	Rhodamine B + fluorescein in the adhesive	CLSM, FESEM, EDX: 2 teeth—2 slices/tooth
(IBa)	Carious Dentine + Riva Light Cure	Rhodamine B + fluorescein in pulp chamber	CLSM, FESEM, EDX: 2 teeth—2 slices/tooth Microhardness: 2 teeth—1 slice/tooth
(IIBa)	Carious Dentine + Riva Self Cure HV	Rhodamine B + fluorescein in pulp chamber	CLSM, FESEM, EDX: 2 teeth—2 slices/tooth Microhardness: 2 teeth—1 slice/tooth
(IIIBa)	Carious Dentine + Activa BioActive Restorative ™	Rhodamine B + fluorescein in pulp chamber	CLSM, FESEM, EDX: 2 teeth—2 slices/tooth Microhardness: 2 teeth—1 slice/tooth
(IVBa)	Carious Dentine + GrandioSO^®^	Rhodamine B + fluorescein in pulp chamber	CLSM, FESEM, EDX: 2 teeth—2 slices/tooth Microhardness: 2 teeth—1 slice/tooth
(IIIBb)	Carious Dentine + Activa BioActive Restorative ™	Rhodamine B + fluorescein in the adhesive	CLSM, FESEM, EDX: 2 teeth—2 slices/tooth
(IVBb)	Carious Dentine + GrandioSO^®^	Rhodamine B + fluorescein in the adhesive	CLSM, FESEM, EDX: 2 teeth—2 slices/tooth

CLSM: confocal laser scanning microscopy; FESEM: field emission scanning electron microscopy; EDX: energy dispersive X-ray.

**Table 2 materials-18-03984-t002:** Percentage concentration by weight of the chemical elements present in the Prime&Bond Active^TM^ adhesive.

C	O	N	Ca	P	Mg
71.92 ± 1.62	17.91 ± 2.16	6.92 ± 1.44	1.78 ± 0.43	1.67 ± 0.09	0.40 ± 0.11

C, carbon; O, oxygen; N, nitrogen; Ca, calcium; P, phosphorus; Mg, magnesium.

**Table 3 materials-18-03984-t003:** Percentage concentration by weight of the chemical elements present in each of the restorative materials used in the study.

	O	C	Si	Al	Sr	F	Na	Ca	Ba
Riva Light Cure	30.32 ± 1.40	25.47 ± 0.65	11.63 ± 0.93	9.75 ± 0.74	12.03 ± 1.41	6.56 ± 0.68	0.81 ± 0.14	0.85 ± 0.17	-
Riva Self Cure HV	31.63 ± 2.48	27.87 ± 6.39	9.96 ± 0.65	9.64 ± 1.51	10.75 ± 1.31	5.83 ± 1.26	0.72 ± 0.14	1.44 ± 0.58	-
Activa BioActive Restorative™	28.60 ± 1.67 ^a^	44.58 ± 3.17 ^a,b,c^	10.61 ± 0.92 ^a^	2.55 ± 0.39 ^b,c^	-	2.08 ± 0.51 ^b,c^	0.31 ± 0.07 ^b,c^	1.93 ± 0.62 ^b^	8.32 ± 0.84
GrandioSO^®^	34.88 ± 2.15 ^b^	22.11 ± 3.34	23.00 ± 2.02 ^c^	3.43 ± 0.20 ^b,c^	-	-	-	-	15.49 ± 0.91 ^a^

HV: high-viscosity; O: oxygen; C: carbon; Si: silicon; Al: aluminium; Sr: strontium; F: fluorine; Na: sodium; Ca: calcium; Ba: barium; Statistical analysis: ANOVA + Tukey’s test (O, C, Al) and Kruskal–Wallis + Dwass–Steel–Critchlow–Fligner test (Si, Na, Ca, Sr, F, Ba). ^a^: *p* < 0.05 vs. GrandioSO^®^; ^b^: *p* < 0.05 vs. Riva Light Cure; ^c^: *p* < 0.05 vs. Riva Self Cure HV.

**Table 4 materials-18-03984-t004:** Percentage concentration by weight of the chemical elements present in the sound dentine near the bonding interface with the different restorative materials.

	O	Ca	C	P	N	Mg	Na	F
Riva Light Cure	36.53 ± 1.49	26.61 ± 1.24	20.39 ± 0.91	12.94 ± 0.36	3.05 ± 0.31	0.77 ± 0.05	0.48 ± 0.05	0.13 ± 0.20
Riva Self Cure HV	33.73 ± 3.09	20.95 ± 1.25	28.17 ± 9.36	10.23 ± 2.50	5.25 ± 1.99 ^a,b^	0.55 ± 0.18 ^b^	0.41 ± 0.21	0.66 ± 0.26 ^b^
Activa BioActive Restorative™	36.54 ± 0.83	26.20 ± 0.93	20.34 ± 1.43	12.68 ± 0.47	3.40 ± 0.30 ^c^	0.63 ± 0.05 ^b^	0.67 ± 0.11 ^b,c^	0.26 ± 0.47
GrandioSO^®^	35.82 ± 1.11	26.27± 1.23	21.95 ± 3.09	12.58 ± 0.59	3.09 ± 0.45	0.63 ± 0.09 ^b^	0.60 ± 0.07 ^c^	-

HV: high-viscosity; O: oxygen; Ca: calcium; C: carbon; P: phosphorus; N: nitrogen; Mg: magnesium; Na: sodium; F: fluorine. Statistical analysis: ANOVA + Tukey’s test (O, Ca, Na, Mg) and Kruskal–Wallis + Dwass–Steel–Critchlow–Fligner test (C, N, F, P). ^a^: *p* < 0.05 vs. GrandioSO^®^; ^b^: *p* < 0.05 vs. Riva Light Cure; ^c^: *p* < 0.05 vs. Riva Self Cure HV.

**Table 5 materials-18-03984-t005:** Percentage concentration by weight of the chemical elements present in the carious dentine near the bonding interface with the different restorative materials.

	O	Ca	C	P	N	Mg	Na	F	Si	Sr	S
Riva Light Cure	36.12 ± 0.10	21.74 ± 2.68	24.34 ± 2.70	10.63 ± 1.17	4.94 ± 0.67	-	0.50 ± 0.08	0.72 ± 0.08	-	0.75 ± 037	-
Riva Self Cure HV	32.45 ± 4.01	23.17 ± 2.11	26.44 ± 3.97	11.20 ± 1.12	4.78 ± 0.76	0.24 ± 0.03	0.37 ± 0.08	0.83 ± 0.30	1.46 ± 0.88	0.51 ± 0.54	-
Activa BioActive Restorative™	32.02 ± 4.05	24.13 ± 4.56	26.04 ± 5.87	11.97 ± 2.45	5.75 ± 1.61	0.30 ± 0.07	0.43 ± 0.09	0.46 ± 0.18 ^c^	0.32 ± 0.15	0.07 ± 0.10 ^b^	0.57 ± 0.28
GrandioSO^®^	33.24 ± 1.59	26.62 ± 1.60	23.81 ± 2.86	12.93 ± 0.76	4.32 ± 0.58	0.41 ± 0.12	0.55 ± 0.12	0.35 ± 0.02 ^c^	0.65 ± 0.24	0.11 ± 0.12	0.32 ± 0.03

HV: high-viscosity; O: oxygen; Ca: calcium; C: carbon; P: phosphorus; N: nitrogen; Mg: magnesium; Na: sodium; F: fluorine; Si: silicon; Sr: strontium; S: sulphur. Statistical analysis: ANOVA + Tukey’s test (O, P, N, Na, F) and the Kruskal–Wallis + Dwass–Steel–Critchlow–Fligner test (Ca, C, Sr, Mg, S, Si). ^b^: *p* < 0.05 vs. Riva Light Cure; ^c^: *p* < 0.05 vs. Riva Self Cure HV.

**Table 6 materials-18-03984-t006:** Microhardness values at the material–sound dentine and carious dentine interface. Microhardness of the materials.

	Materials	Material-Sound Dentine Interface	Material-Carious Dentine Interface	t-Test *
Riva Light Cure	62.51 ± 6.23 VHN ^c^	52.78 ± 9.79 VHN ^c^	13.03 ± 2.98 * VHN ^c^	*p* < 0.001
Riva Self Cure HV	82.50 ± 16.80 VHN	36.33 ± 6.23 VHN	34.56 ± 4.31 VHN	*p* = 0.605
Activa BioActive Restorative™	33.27 ± 8.68 VHN ^a,b,c^	32.97 ± 3.99 VHN ^a,b^	16.98 ± 4.98 * VHN ^c^	*p* < 0.001
GrandioSO^®^	111.04 ± 11.13 VHN ^b,c^	56.87 ± 8.95 VHN ^c^	18.62 ± 11.71 * VHN ^c^	*p* < 0.001

HV: high-viscosity; DS: healthy dentine; DC: carious dentine; Statistical analysis between materials: ANOVA + Tukey’s test. ^a^: *p* < 0.05 vs. GrandioSO^®^; ^b^: *p* < 0.05 vs. Riva Light Cure; ^c^: *p* < 0.05 vs. Riva Self Cure HV. * Differences in microhardness between the material–sound dentine and material–carious dentine interfaces.

## Data Availability

The original contributions presented in this study are included in the article. Further inquiries can be directed to the corresponding author.

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
