# Peer review of "Determination of the Degree of Penetration of Glass Ionomer Cements in the Healthy and Decayed Dentine of Permanent Molars"

_materials, 2025, doi:10.3390/ma18173984_

Round 1

Reviewer 1 Report

Comments and Suggestions for Authors

This study provides a valuable comparison of the interfacial properties of three restorative materials in sound versus carious dentin using multiple analytical techniques. The experimental design is generally sound, and the dataset is substantial. However, the following issues require specific attention:

1. Image readability is poor and need to be improved. Annotation labels and some scale bars are too small. A descriptive caption beyond simple symbol definitions is mandatory for each figure.
2. The rationale for employing two distinct methods for fluorescein introduction within the study design is unclear. Please clarify whether this methodological difference could introduce variability or bias in the measured penetration depth results. 
3. The use of a 300g load for microhardness testing appears higher than common protocols for dental biomaterials. Please provide evidence (such as representative optical micrographs or SEM images) to confirm that this load did not cause unreliable measurements due to cracking or deformation around the indentation sites.
4. Please more explicitly summarize the study's new findings and discuss the possible clinical implications.
5. The limitations stemming from the modest sample size and the exclusive use of an in vitro model should be acknowledged to highlight the necessity for future research.

Author Response

Comments and Suggestions for Authors 1. Image readability is poor and need to be improved. Annotation labels and some scale bars are too small. A descriptive caption beyond simple symbol definitions is mandatory for each figure.

Response: A document with larger images and symbols is attached, for which Figure 1 has had to be divided into 1A, 1B and 1C and Figure 2 into 2A and 2B.

Therefore, make the following changes:

Page 8, line 292: CLSM and FESEM images are shown for the healthy (Figure 1A-C) and carious (Figure 2A-B) dentine groups.

Page 14, paragraph 1, line 421: (Figure. 1B, CLSM group IIIAb).

Page 14, paragraph 3, line 426: (Figure 1A)

Page 14, paragraph 4, line 433: (Figure 2A)

Page 14, paragraph 6, línea 451: (Figure 1B and 1C, respectively 1, groups IIIA and IVA, respectively)

Page 14, paragraph 6, line 452: (Figure 2B, groups IIIB and IVB, respectively)

Page 15, paragraph 4, line 487: (Figure 2B, CLSM IVBa)

Comments and Suggestions for Authors 2. The rationale for employing two distinct methods for fluorescein introduction within the study design is unclear. Please clarify whether this methodological difference could introduce variability or bias in the measured penetration depth results.

Response: page 14, paragraph 1

The use of fluorochromes does not have a standardised protocol. In this study, we used two different systems in order to observe whether the use of one or other system influences the interpretation of the results; on the one hand, the introduction of fluorescein through the pulp chamber for 3 hours [18,19], and on the other hand, the combination with rhodamine B, mixed with each restorative material [23,26], which facilitates the visualisation of the tooth structure and the restorative material, reducing possible misinterpretations, and is, therefore, widely used in CLSM [31]. In groups III and IV, a subgroup b was also added where fluorescein was mixed only with the adhesive [31], which significantly improved the visualisation of the interphase morphology, allowing the tags and intertubular connections to be observed more clearly (Figure. 1B, CLSM group IIIAb). Since there may be interpretative alterations, it is highly recommended to continue with comparative studies between the different fluorochrome incorporation systems to establish a standardised methodology that will lead to homogeneous interpretations of the results.

Comments and Suggestions for Authors 3. The use of a 300g load for microhardness testing appears higher than common protocols for dental biomaterials. Please provide evidence (such as representative optical micrographs or SEM images) to confirm that this load did not cause unreliable measurements due to cracking or deformation around the indentation sites.

Response:

The samples that were subjected to the microhardness test were not photographed or subsequently observed in FESEM, so we cannot attach any micrographs. However, when we perform the measurements, we directly observe the indentations to determine the length of the diameters and in no case do we observe any cracks or deformations at the indentation sites.

Comments and Suggestions for Authors 4. Please more explicitly summarize the study's new findings and discuss the possible clinical implications.

Response: insert before paragraph 8, page 15

Taking into account the results obtained in this investigation at morphological, chemical and mechanical levels, it could be considered that clinically, in healthy dentine, none of the three materials stood out above the rest in all the parameters analysed. The bioactive resin together with the adhesive showed very good adaptation at the interface with healthy dentine, but it was the one that obtained the lowest microhardness values; however, the glass ionomers showed good bonding and better results in microhardness, especially the RM GIC. On the other hand, in carious dentine, the ionomers were more stable with respect to their interfacial hybridisation and HV GIC obtained the highest microhardness value, in contrast to the bioactive resin with the adhesive, which showed the lowest microhardness record and very poor bonding. All materials obtained higher fluoride concentrations in carious dentine than in healthy dentine.

Considering the OMI philosophy, which urges the realisation of our restorations on demineralised dentine substrate, HV GIC could be a material with good behaviour as it presents the best combination of adhesion and microhardness, as well as possessing ions that favour the remineralisation process.

Comments and Suggestions for Authors 5. The limitations stemming from the modest sample size and the exclusive use of an in vitro model should be acknowledged to highlight the necessity for future research

Response: Page 15, paragraph 8:

This work has certain limitations that should be considered when interpreting the results. Firstly, the method for fluorochrome application is not standardised, which may introduce variability in the visualisation of the interfaces. In addition, variability in the depth and extent of carious lesions in the samples could influence the adhesion and microhardness obtained, limiting the generalisability of the findings. On the other hand, the incorporation of adhesive in some groups could have affected the potential biological activity of materials classified as bioactive, preventing the complete evaluation of this property. Finally, because the sample size is not very large the analyses were performed in vitro, so the results could differ in real clinical conditions, where additional environmental and biological factors influence the behaviour of restorative materials, further research is needed to establish more accurate conclusions.

Reviewer 2 Report

Comments and Suggestions for Authors

This manuscript presents a well-designed in vitro study evaluating the morphological, chemical, and mechanical interaction of three restorative materials with healthy and carious dentine. The study uses robust methodology (CLSM, FESEM, EDX, and Vickers microhardness) and offers clinically relevant findings for minimally invasive restorative dentistry. However, the manuscript requires revision to improve clarity, completeness, and consistency, particularly in the methods, discussion, and presentation of figures/tables.

Abstract

Include the number of teeth per group to clarify sample size.

Introduction

Add more context on why Activa BioActive Restorative™ is controversial (bioactivity vs. mechanical properties).

Clarify the distinction between "bioactive" and "adhesive" behavior.

Materials and Methods

include more detail on sample size calculation.

Explain how examiner calibration was managed during imaging interpretation.

Define acronyms on first use in each section (e.g., CLSM, FESEM).

Add more detail to the statistical analysis: was normality confirmed per group? How were missing values handled?

Results

Clarify which differences were statistically significant in each subsection—some statements are vague (e.g., "better bonding observed").

Add clarity to Table 6 (Vickers Microhardness): bold significant values, add footnotes.

Discussion

Avoid over-citation of individual studies in a single paragraph (e.g., references [18, 23, 26, 31]).

Consider summarizing key findings more directly about each material.

Add a separate short paragraph on clinical implications and recommendations for practice.

Expand on the limitations (e.g., “differences in lesion depth” should be quantified or illustrated if possible).

Author Response

  • Abstract

Include the number of teeth per group to clarify sample size.

Line 29:

protocolos and subsequently divided into blades. So, twenty-four samples were uses for each group (sound and carious dentin) for interface analysis using confocal laser scanning microscopy, field emisión scanning electron microscop and energy dispersive X-ray spectroscopy and, another eight simples were used for each group (sound and carious dentin) for Vickers microhardness testing.

  • Introduction
  1. Add more context on why Activa BioActive Restorative™is controversial (bioactivity vs. mechanical properties).
  2. Clarify the distinction between "bioactive" and "adhesive" behavior.

Suggestions for points 1 and 2; Inser after paragraph 2, page 3:

However, there are discrepances between the manufacturer’s claims and the scientific literature, which states that the bioactivity of a materials is its ability to induce a biological response, which, to date, has not been demonstrated; furthermore, the adhesive system used to improve the mechanical properties of Activa BioActiv Restaurative™, may act as a barrier between the material and the tooth tissue, possibly limiting or preventing the bioactive effect by preventing direct contact between the two [17,34,35].

  • Materials and Methods
  1. Include more detail on sample size calculation (same response to review 3)

The limited sample size results from the challenges associated with obtaining human teeth that meet both the quantity and quality requirements for research. These challenges include heightened awareness of infection risks and ethical concerns, the fact that many teeth are extracted due to extensive caries, lesions, or other structural defects, and difficulties in controlling variables such as the origin and age of the specimens, which complicate the establishment of sample homogeneity. [Teruel Jde D, Alcolea A, Hernández A, Ruiz AJ. Comparison of chemical composition of enamel and dentine in human, bovine, porcine and ovine teeth. Arch Oral Biol. 2015 May;60(5):768-75. doi: 10.1016/j.archoralbio.2015.01.014]

If the reviewer and editor consider it appropriate, we are willing to include the term "pilot study" in the title to better reflect the scope and limitations of the study.

  1. Explain how examiner calibration was managed during imaging interpretation.

The authors have substantial experience in the application of this methodology and the interpretation of the resulting images. Below is a selection of published studies by our group employing these imaging techniques, including confocal laser microscopy, SEM, and FESEM.

  1. Toledano M, Osorio R, Osorio E, Cabello I, Toledano-Osorio M, Aguilera FS. In vitro mechanical stimulation facilitates stress dissipation and sealing ability at the conventional glass ionomer cement-dentin interface. J Dent. 2018 Jun;73:61-69. doi: 10.1016/j.jdent.2018.04.006. PMID: 29653140.

  1. Toledano M, Osorio R, Osorio E, Cabello I, Toledano-Osorio M, Aguilera FS. A zinc chloride-doped adhesive facilitates sealing at the dentin interface: A confocal laser microscopy study. J Mech Behav Biomed Mater. 2017 Oct;74:35-42. doi: 10.1016/j.jmbbm.2017.04.030. PMID: 28535395.

  1. Toledano M, Osorio R, Cabello I, Osorio E, Toledano-Osorio M, Aguilera FS. Oral Function Improves Interfacial Integrity and Sealing Ability Between Conventional Glass Ionomer Cements and Dentin. Microsc Microanal. 2017 Feb;23(1):131-144. doi: 10.1017/S1431927617000010. PMID: 28148310.

  1. Osorio R, Cabello I, Medina-Castillo AL, Osorio E, Toledano M. Zinc-modified nanopolymers improve the quality of resin-dentin bonded interfaces. Clin Oral Investig. 2016 Dec;20(9):2411-2420. doi: 10.1007/s00784-016-1738-y. PMID: 26832781.

  1. Toledano M, Osorio E, Cabello I, Aguilera FS, López-López MT, Toledano-Osorio M, Osorio R. Nanoscopic dynamic mechanical analysis of resin-infiltrated dentine, under in vitro chewing and bruxism events. J Mech Behav Biomed Mater. 2016 Feb;54:33-47. doi: 10.1016/j.jmbbm.2015.09.003. PMID: 26414515.

  1. Toledano M, Cabello I, Aguilera FS, Osorio E, Toledano-Osorio M, Osorio R. Improved Sealing and Remineralization at the Resin-Dentin Interface After Phosphoric Acid Etching and Load Cycling. Microsc Microanal. 2015 Dec;21(6):1530-1548. doi: 10.1017/S1431927615015317. PMID: 26471836.

  1. Toledano M, Aguilera FS, Osorio E, Cabello I, Osorio R. Microanalysis of thermal-induced changes at the resin-dentin interface. Microsc Microanal. 2014 Aug;20(4):1218-33. doi: 10.1017/S1431927614000944. PMID: 24905087.

  1. Toledano M, Aguilera FS, Sauro S, Cabello I, Osorio E, Osorio R. Load cycling enhances bioactivity at the resin-dentin interface. Dent Mater. 2014 Jul;30(7):e169-88. doi: 10.1016/j.dental.2014.02.009. PMID: 24630703.

  1. Toledano M, Osorio E, Aguilera FS, Sauro S, Cabello I, Osorio R. In vitro mechanical stimulation promoted remineralization at the resin/dentin interface. J Mech Behav Biomed Mater. 2014 Feb;30:61-74. doi: 10.1016/j.jmbbm.2013.10.018. PMID: 24246198.

  1. Toledano M, Sauro S, Cabello I, Watson T, Osorio R. A Zn-doped etch-and-rinse adhesive may improve the mechanical properties and the integrity at the bonded-dentin interface. Dent Mater. 2013 Aug;29(8):e142-52. doi: 10.1016/j.dental.2013.04.024. PMID: 23764024.

  1. Ortiz-Ruiz AJ, Martínez-Marco JF, Pérez-Silva A, Serna-Muñoz C, Cabello I, Banerjee A. Influence of Fluoride Varnish Application on Enamel Adhesion of a Universal Adhesive. J Adhes Dent. 2021;23(1):47-56. doi: 10.3290/j.jad.b916831. PMID: 33512115.

  1. Vicente A, Ortiz-Ruiz AJ, González-Paz BM, Martínez-Beneyto Y, Bravo-González LA. Effectiveness of a toothpaste and a serum containing calcium silicate on protecting the enamel after interproximal reduction against demineralization. Sci Rep. 2021 Jan 12;11(1):834. doi: 10.1038/s41598-020-80844-7. PMID: 33437013; PMCID: PMC7804454.

  1. Ortiz-Ruiz AJ, Muñoz-Gómez IJ, Pérez-Pardo A, Germán-Cecilia C, Martínez-Beneyto Y, Vicente A. Influence of fluoride varnish on shear bond strength of a universal adhesive on intact and demineralized enamel. 2018 Oct;106(4):460-468. doi: 10.1007/s10266-018-0363-4. PMID: 29704075.

  1. Serna Muñoz C, Pérez Silva A, Solano F, Castells MT, Vicente A, Ortiz Ruiz AJ. Effect of antibiotics and NSAIDs on cyclooxygenase-2 in the enamel mineralization. Sci Rep. 2018 Mar 7;8(1):4132. doi: 10.1038/s41598-018-22607-z. PMID: 29515175; PMCID: PMC5841276.

  1. Vicente A, Ortiz Ruiz AJ, García López M, Martínez Beneyto Y, Bravo-González LA. Enamel Resistance to Demineralization After Bracket Debonding Using Fluoride Varnish. Sci Rep. 2017 Nov 9;7(1):15183. doi: 10.1038/s41598-017-15600-5. PMID: 29123323; PMCID: PMC5680324.

  1. Vicente A, Ortiz Ruiz AJ, González Paz BM, García López J, Bravo-González LA. Efficacy of fluoride varnishes for preventing enamel demineralization after interproximal enamel reduction. Qualitative and quantitative evaluation. PLoS One. 2017 Apr 21;12(4):e0176389. doi: 10.1371/journal.pone.0176389. PMID: 28430810; PMCID: PMC5400240.

  1. Ortiz Ruiz AJ, Vicente A, Camacho Alonso F, López Jornet P. A new use for self-etching resin adhesives: cementing bone fragments. J Dent. 2010 Sep;38(9):750-6. doi: 10.1016/j.jdent.2010.06.002. PMID: 20600553.

  1. Define acronyms on first use in each section (e.g., CLSM, FESEM).

Added footnote below the Table 1. Grupos experimentales:

CLSM: Confocal Laser Scanning Microscopy; FESEM: Field Emission Scanning Electron Microscopy; EDX: Energy Dispersive X-Ray

Added to the manuscript (Page 7, paragraph 2, line 240):

CLSM (Confocal Laser Scanning Microscopy), FESEM: (Field Emission Scanning Electron Microscopy) and EDX (Energy Dispersive X-Ray)

  1. Add more detail to the statistical analysis: was normality confirmed per group? How were missing values handled?

We have written a new statistical analysis section (page 8, line 283), where everything done is explained in detail.

Jamovi v2.3 was used for statistical analysis. Normality was assessed with Shapiro-Wilk and homogeneity with Levene. To compare elemental compositions, ANOVA or Kruskal-Wallis were applied as appropriate, followed by post-hoc tests. Microhardness was analysed by ANOVA and Tukey's test or T-test when groups were unified. Significant differences were considered significant at p < 0.05.

The variables analyzed in this study included: the weight percentage concentration of chemical elements present in the adhesive; the weight percentage concentration of elements in each restorative material; and the elemental composition of sound and carious dentin adjacent to the adhesive interface. Additionally, microhardness was assessed at the material–dentin interface (both sound and carious), within the restorative materials, and in sound and carious dentin themselves.

Statistical analyses were performed using Jamovi version 2.3. Descriptive statistics were computed for all study variables. Shapiro–Wilk tests were used to assess normality, and Levene’s test was applied to evaluate homogeneity of variances. To assess differences in material composition, a one-way ANOVA was conducted followed by Tukey’s post hoc test when assumptions of normality and homoscedasticity were met (elements: O, C, Al). For elements that did not meet these assumptions (Si, Na, Ca, Sr, F, Ba), the Kruskal–Wallis test followed by the Dwass–Steel–Critchlow–Fligner test was used.

To examine differences in dentin composition near the adhesive interface, the same approach was applied. A one-way ANOVA with Tukey’s post hoc test was used when assumptions were satisfied (sound dentin: O, Ca, Na, Mg; carious dentin: O, P, N, Na, F), and the Kruskal–Wallis test with Dwass–Steel–Critchlow–Fligner post hoc analysis was used when assumptions were violated (sound dentin: C, N, F, P; carious dentin: Ca, C, Sr, Mg, S, Si).

As microhardness values across the experimental groups met the assumptions of normality and homogeneity of variance, a one-way ANOVA followed by Tukey’s test was used to identify pairwise differences. Differences in microhardness between sound and carious dentin and the material-sound dentin and carious dentin interfaces were evaluated using an independent samples t-test.

A p-value of < 0.05 was considered statistically significant.

  • Results
  1. Clarify which differences were statistically significant in each subsection—some statements are vague (e.g., "better bonding observed").

For the study of the simples under CLSM and FESEM, the analysis of the entire perimeter of the interface was simply observational, which is why no reference is made to statistical differences.. Howewer, statistical analysis wa performed for EDX and Vickers microhardness:

Page 10, line 346:

EDX analysis revealed the common ions mostly present in all materials were oxygen, carbon, silicon and aluminium, with strontium only in Riva Ligth Cure and Riva Self Cur HV, barium in Activa BioActive Restorative™ and GrandioSO®  and fluorine absent in GrandioSO®. However, there were significant differences in the elemental composition concentration of the restorative materials, except between Riva Light Cure and Riva Self Cure HV, which no significant difference was found. In a more specific analysis, there was a statistically higher concentration of carbon in Activa BioActive Restorative™ (p<0.001), aluminium, sodium and fluoride in Riva Ligth Cure and Riva Self Cur HV (p<0.001). showed similar compositions with a prominent presence of strontium, fluoride and aluminium, after the common major ions (oxygen and carbon). In contrast, Activa BioActive Restorative™ had higher carbon and barium content, while GrandioSO® was similar to the latter, but without fluoride. At the interface with healthy dentin, significant differences in nitrogen, magnesium, sodium and fluoride were observed and, in carious dentin, only fluoride and strontium varied significantly between groups. Elemental mapping confirmed these results, showing that the GICs presented a clear distribution of elements such as alumnium, silicon, strontium and fluorine in the material and calcium and phosphorus in the dentine, while Activa and GrandioSO® were dominated by barium, a high concentration of carbon in the adhesive and no fluorine in GrandioSO®, although it was detected in areas of carious dentine. All results can be seen in the tables (Table 2,3,4 and 5).

  1. Add clarity to Table 6 (Vickers Microhardness): bold significant values, add footnotes.

Signiticant values are highlighted in bold and footnotes have been added. In addition, other modifications have been made (suggestions reviewer 3)

Table 6. Microhardness values at the material-sound dentin and carious dentin interface. Microhardness of the materials

MICRODURE

Materials

Material-sound dentin interface

Material- carious dentin interface

T-test*

Riva Light Cure

62.51 ± 6.23 VHN

c

52.78 ± 9.79 VHN c

13.03 ± 2.98 * VHN

c

p<0.001

Riva Self Cure HV

82.50 ± 16.80 VHN

36.33 ± 6.23 VHN

34.56 ± 4.31 VHN

p=0.605

Activa BioActive Restorative™

33.27 ± 8.68 VHN

a, b, c

32.97 ± 3.99 VHN a, b

16.98 ± 4.98 * VHN

c

p<0.001

GrandioSO®

111.04 ± 11.13 VHN b, c

56.87 ± 8.95 VHN c

18.62 ± 11.71 * VHN c

p<0.001

HV: High Viscosity; DS: Healthy Dentin; DC: Carious Dentin; Statistical analysis between materials: ANOVA + Tukey's test. a: p<0.05 vs. GrandioSO®; b: p<0.05 vs Riva Light Cure; c: p<0.05 vs. Riva Self Cure HV. * Differences in microhardness between the material-sound dentin and material-carious dentin interfaces.

  • Discussion
  1. Avoid over-citation of individual studies in a single paragraph (e.g., references [18, 23, 26, 31]).

Thank you for your recommendation.

  1. Consider summarizing key findings more directly about each material.
  2. Add a separate short paragraph on clinical implications and recommendations for practice.

Suggestions for points 2 and 3 have also been requested by Reviewer 1. Inser before paragraph 8, page 15:

Taking into account the results obtained in this investigation at morphological, chemical and mechanical levels, it could be considered that clinically, in healthy dentine, none of the three materials stood out above the rest in all the parameters analysed. The bioactive resin together with the adhesive showed very good adaptation at the interface with healthy dentine, but it was the one that obtained the lowest microhardness values; however, the glass ionomers showed good bonding and better results in microhardness, especially the RM GIC. On the other hand, in carious dentine, the ionomers were more stable with respect to their interfacial hybridisation and HV GIC obtained the highest microhardness value, in contrast to the bioactive resin with the adhesive, which showed the lowest microhardness record and very poor bonding. All materials obtained higher fluoride concentrations in carious dentine than in healthy dentine.

Considering the OMI philosophy, which urges the realisation of our restorations on demineralised dentine substrate, HV GIC could be a material with good behaviour as it presents the best combination of adhesion and microhardness, as well as possessing ions that favour the remineralisation process.

  1. Expand on the limitations (e.g., “differences in lesion depth” should be quantified or illustrated if possible).

The manuscript already mentions the differences in the depth of carious lesions between the different simples are a limiration in this study.

Page 15, paragraph 8, line 509:

This work has certain limitations …. In addition, variability in the depth and extent of carious lesions in the samples could influence the adhesion and microhardness obtained, limiting the generalisability of the findings.

Page 11, Figure 2:

The CLSM image of group IVBa is the example of less interfacial mismatch when carious dentin was scarcer then in the other simples with greater amounts of carious dentin.

Reviewer 3 Report

Comments and Suggestions for Authors

The article is very interesting and boldly written. The topic is very interesting, as maintaining a proper bond between the tooth and the restorative material over an extended period of time is a significant clinical challenge.

You could perform the second part of your research, after the teeth have been stored for a longer period, as the tooth gradually deteriorates and the tooth-material bond changes.
I have a few comments:

Abstract
It would be good to add numerical values ​​to the obtained results. This often attracts the reader's attention more.

Introduction
line 61
In this scenario, composite resins would need to be formulated slightly differently. After all, composite materials are not commonly used in the remineralization process solely due to their high mechanical and aesthetic properties. Line 91
One of the most representative examples is Activa BioActive Restorative™ (Pulpdent Corp., USA), a material that does not contain BPA or its derivatives, and whose composition includes a bioactive resin matrix reinforced with silanized fillers and a dual activation (chemical and photonic) – with this name in bold and the description, it looks like surreptitious advertising. To be fair, add 2-3 days about the other GiC you're testing, namely Riva.

And what thesis will you put forward at the beginning of your research?

M&M
ISO/TS 11405:2015 - Testing of adhesion to tooth structure - please add this as a reference, thank you.
Table 1. Abbreviations for the techniques used in the tests should be explained directly below the table so that it is self-explanatory when extracted from the text. Isn't a sample of two teeth examined in one research group too small? Please explain?

Instead, in the introduction, you mention the GiC intended for testing, the same in the abstract, but in Table 1, the composite material Grandio appears?

Line 190
D truncated conical diamond bur - what shape, what brand, what contra-angle rotation?

Line 199
Rhodamine B 0.05% by weight for subsequent visualization (RhB; Sigma-Aldrich Chemie Gmbh, Riedstr., Germany) {Citation} was added to the mixture and spatulated manually - citation? added to the mixture of glass ionomer cement. Line 202
Riva Self Cure 202
HV was applied in a single increment, and after curing, Riva Coat was applied and cured. - new thought, new line
line 2025
Prime&Bond Active™ - manufacturer? Why did you use a bonding system from a manufacturer other than Pulpdent?

Results

Nice photos - congratulations
The table captions are missing. There's number 2 and then number 6. Are the numerical values ​​in tables 2, 3, 4, and 5 in [%]?
line 398 under table
p<0.05 a vs. GRANDIOSO; b vs. RLC; c vs. RSC - missing, and it would be good to explain what is statistically significant relative to what. Maybe 1 sentence?
line 402
; p<0.05 a vs. GRANDIOSO; b vs. RLC; c vs. RSC - in Table 6, the p-level is 0.001 or p=0.605, and in the description, p<0.05? The p-level of 0.605 is not statistically significant. Table 6 lacks units defining the Vickers hardness scale.

Discussion
line 432
[7,25,28,32+] - what does 32+ mean as a ref?
Line 456 - a bold claim about bioactive materials, but true. Congratulations! According to the new guidelines of the MDR directive in the EU, the manufacturer of the material must submit irrefutable evidence for all its claims regarding medical materials.

Good luck with your further research!

Author Response

  • Abstract

It would be good to add numerical values ​​to the obtained results. This often attracts the reader's attention more.

Line 35

Elemental analysis revealed similar compositions among materials, with no significant differents in material concentrations among the ionomers, while there were significant differences with the others materials. On the other hand, some variations were observed in  the sulphur, fluoride and strontium content depending on dentine condition. Microhardness values were higher in healthy dentine  than in carious dentin for all materials (p<0.001), except the high-viscosity glass ionomer, which maintained stable hardness in both substrates (36.33 ± 6.23 VHN vs 34.56 ± 4.31 VHN; p=0.605).

  • Introduction
  1. In this scenario, composite resins would need to be formulated slightly differently. After all, composite materials are not commonly used in the remineralization process solely due to their high mechanical and aesthetic properties.

We agree with the reviewer. The paragraph may be omitted without compromising the overall clarity or intent of the introduction.

  1. Line 91
    One of the most representative examples is Activa BioActive Restorative™ (Pulpdent Corp., USA), a material that does not contain BPA or its derivatives, and whose composition includes a bioactive resin matrix reinforced with silanized fillers and a dual activation (chemical and photonic) – with this name in bold and the description, it looks like surreptitious advertising. To be fair, add 2-3 days about the other GiC you're testing, namely Riva.

We have removed the bold highlightinf of Activa BioActive Restorative™ and we added (after of line 90) a few senteces about Riva ionomers:

According to the manufacturer, Riva glass ionomer cements (SDI) are composed of multiple ultrafine particles of varying sizes, which ensure restorations of good strength and aesthetics, as well as easy handling (no etching or adhesive is required, as they provide chemical bonding). In addition, they contain fluoride and strontium ions that improve the bio-mineralisation of tooth tissue.

  1. And what thesis will you put forward at the beginning of your research?

Insert after of line 102, before the objetive:

In this investigation, it is therefore a null hypothesis that the mechanical behaviour of the carious dentine - resin-modified glass ionomer and bioactive resin interfaces is no better than when the glass ionomer is a conventional high viscosity glass ionomer.

In response to this hypothesis, insert after line 533:

Finally, the null hypothesis is accepted, as the mechanical behavior of the carious dentin-high-viscosity glass ionomer interface is better than that of resin-modified glass ionomer and bioactive resin.

  • M&M
  1. ISO/TS 11405:2015 - Testing of adhesion to tooth structure - please add this as a reference, thank you.

Line 121:

months after extraction, according to ISO TS 11405:2015-Testing of adhesion to tooth structure.

  1. Suggestion made about Table 1. Abbreviations for the techniques used in the tests should be explained directly below the table so that it is self-explanatory when extracted from the text.

CLSM: Confocal Laser Scanning Microscopy; FESEM: Field Emission Scanning Electron Microscopy; EDX: Energy Dispersive X-Ray.

Isn't a sample of two teeth examined in one research group too small? Please explain? (same response to review 2)

The limited sample size results from the challenges associated with obtaining human teeth that meet both the quantity and quality requirements for research. These challenges include heightened awareness of infection risks and ethical concerns, the fact that many teeth are extracted due to extensive caries, lesions, or other structural defects, and difficulties in controlling variables such as the origin and age of the specimens, which complicate the establishment of sample homogeneity. [Teruel Jde D, Alcolea A, Hernández A, Ruiz AJ. Comparison of chemical composition of enamel and dentine in human, bovine, porcine and ovine teeth. Arch Oral Biol. 2015 May;60(5):768-75. doi: 10.1016/j.archoralbio.2015.01.014]

If the reviewer and editor consider it appropriate, we are willing to include the term "pilot study" in the title to better reflect the scope and limitations of the study.

  1. Instead, in the introduction, you mention the GiC intended for testing, the same in the abstract, but in Table 1, the composite material Grandio appears?

GrandioSO composite resin has been used as a control group, which is why it appears in the tables, but is not mentioned in the abstract,…

  1. Line 190
    D truncated conical diamond bur - what shape, what brand, what contra-angle rotation?

No contra-angle was used to make the cavity.

Now it stated (page 6, line 190):

“… using a tapered bur 6830L.314. 012 (Komet Dental, Gebr. Brasseler GmbH &Co. KG, Lemgo, Germany) in a Synea Vision TK-94 turbine at 360,000 rpm (W&H Dentalwerk GmbH; Salzburg, Austria).”

  1. Line 199
    Rhodamine B 0.05% by weight for subsequent visualization (RhB; Sigma-Aldrich Chemie Gmbh, Riedstr., Germany) {Citation} was added to the mixture and spatulated manually - citation? added to the mixture of glass ionomer cement.

Rhodamine B 0.05% by weight for subsequent visualization (RhB; Sigma-Aldrich Chemie Gmbh, Riedstr., Germany) [23]

  1. Line 202
    Riva Self Cure HVÒ was applied in a single increment, and after curing, Riva CoatÒ was applied and cured. -new thought, new line

…Ultra LED lamp (Kerr, CA, USA), finishing with the application and light curing of Riva CoatÒ .

Riva Self Cure HVÒ was applied in a single increment and after curing, Riva CoatÒ was applied and cured.

  1. line 225
    Prime&Bond Active™ - manufacturer?

Prime&Bond Active™ (Densply Sirona; Charlotte, EE. UU.)

Why did you use a bonding system from a manufacturer other than Pulpdent?

We selected Prime&Bond Active as universal adhesive due to its extensively documented efficacy in the literature and the existing clinical experience of our research group with its use.

  • Results

Nice photos - congratulations
The table captions are missing. There's number 2 and then number 6. Are the numerical values ​​in tables 2, 3, 4, and 5 in [%]?

Line 398 under table
p<0.05 a vs. GRANDIOSO; b vs. RLC; c vs. RSC - missing, and it would be good to explain what is statistically significant relative to what. Maybe 1 sentence?

Line 402
; p<0.05 a vs. GRANDIOSO; b vs. RLC; c vs. RSC - in Table 6, the p-level is 0.001 or p=0.605, and in the description, p<0.05? The p-level of 0.605 is not statistically significant.

Table 6 lacks units defining the Vickers hardness scale.

The reviewer is correct in all of their suggestions. We've changed the table titles and footnotes, added the statistical analyses used, and reworded the statistical significance. We've also modified the statistical analysis section to make it clearer. We have corrected some mistakes detected in the tables and added VHN to the microhardness units in Table 6.

Table 2. Percentage concentration by weight of the chemical elements present in the Prime&Bond activeTM adhesive

ADHESIVE

C

O

N

Ca

P

Mg

71.92 ± 1.62

17.91 ± 2.16

6.92 ± 1.44

1.78 ± 0.43

1.67 ± 0.09

0.40 ± 0.11

C, carbon; O, oxygen; N, nitrogen; Ca, calcium; P, phosphorus; Mg, magnesium.

Table 3. Percentage concentration by weight of the chemical elements present in each of the restorative materials used in the study

MATERIAL

O

C

Si

Al

Sr

F

Na

Ca

Ba

Riva Light Cure

30.32 ± 1.40

25.47 ± 0.65

11.63 ± 0.93

9.75 ± 0.74

12.03 ± 1.41

6.56 ± 0.68

0.81 ± 0.14

0.85 ± 0.17

-

Riva Self Cure HV

31.63 ± 2.48

27.87 ± 6.39

9.96 ± 0.65

9.64 ± 1.51

10.75 ± 1.31

5.83 ± 1.26

0.72 ± 0.14

1.44 ± 0.58

-

Activa BioActive Restorative™

28.60 ± 1.67

a

44.58±  3.17

a, b, c

10.61 ± 0.92

a

2.55±  0.39

b, c

-

2.08±  0.51

b, c

0.31±  0.07

b, c

1.93±  0.62

b

8.32 ± 0.84

GrandioSO®

34.88±  2.15

b

22.11± 3.34

23.00 ± 2.02

c

3.43 ± 0.20

b, c

-

-

-

-

15.49±  0.91

a

HV: High Viscosity; O: oxygen; C: carbon; Si: silicon; Al: aluminium; Sr: strontium; F: fluorine; Na: sodium; Ca: calcium; Ba: barium; Statistical analysis: ANOVA + Tukey's test (O, C, Al) and Kruskal-Wallis + Dwass-Steel-Critchlow-Fligner test (Si, Na, Ca, Sr, F, Ba). a: p<0.05 vs. GrandioSO®; b: p<0.05 vs Riva Light Cure; c: p<0.05 vs. Riva Self Cure HV.

Table 4. Percentage concentration by weight of the chemical elements present in sound dentin near the bonding interface with the different restorative materials.

DENTINA

 SANA

O

Ca

C

P

N

Mg

Na

F

Riva Light Cure

36.53 ± 1.49

26.61 ± 1.24

20.39 ± 0.91

12.94 ± 0.36

3.05 ± 0.31

0.77 ± 0.05

0.48 ± 0.05

0.13 ± 0.20

Riva Self Cure HV

33.73 ± 3.09

20.95 ± 1.25

28.17 ± 9.36

10.23 ± 2.50

5.25 ± 1.99

a, b

0.55 ± 0.18

b

0.41 ± 0.21

0.66 ± 0.26

b

Activa BioActive Restorative™

36.54 ± 0.83

26.20±  0.93

20.34 ± 1.43

12.68±  0.47

3.40 ± 0.30

c

0.63±  0.05

b

0.67±  0.11

b, c

0.26±  0.47

GrandioSO®

35.82±  1.11

26.27± 1.23

21.95 ± 3.09

12.58 ± 0.59

3.09 ± 0.45

0.63 ± 0.09

b

0.60 ± 0.07

c

-

HV: High Viscosity; O: oxygen; Ca: calcium; C: carbon; P: phosphorus; N: nitrogen; Mg: magnesium; Na: sodium; F: fluorine. Statistical analysis: ANOVA + Tukey's test (O, Ca, Na, Mg) and Kruskal-Wallis + Dwass-Steel-Critchlow-Fligner test (C, N, F, P). a: p<0.05 vs. GrandioSO®; b: p<0.05 vs Riva Light Cure; c: p<0.05 vs. Riva Self Cure HV.

Table 5. Percentage concentration by weight of the chemical elements present in carious dentin near the bonding interface with the different restorative materials.

DENTINA

CARIADA

O

Ca

C

P

N

Mg

Na

F

Si

Sr

S

Riva Light Cure

36.12 ± 0.10

21.74 ± 2.68

24.34 ± 2.70

10.63 ± 1.17

4.94 ± 0.67

-

0.50 ± 0.08

0.72 ± 0.08

-

0.75 ± 037

-

Riva Self Cure HV

32.45 ± 4.01

23.17 ± 2.11

26.44 ± 3.97

11.20 ± 1.12

4.78 ± 0.76

0.24 ± 0.03

0.37 ± 0.08

0.83 ± 0.30

1.46 ± 0.88

0.51 ± 0.54

-

Activa BioActive Restorative™

32.02 ± 4.05

24.13 ± 4.56

26.04 ± 5.87

11.97 ± 2.45

5.75 ± 1.61

0.30 ± 0.07

0.43 ± 0.09

0.46 ± 0.18

c

0.32 ± 0.15

0.07 ± 0.10

b

0.57 ± 0.28

GrandioSO®

33.24 ± 1.59

26.62 ± 1.60

23.81 ± 2.86

12.93 ± 0.76

4.32 ± 0.58

0.41 ± 0.12

0.55 ± 0.12

0.35 ± 0.02

c

0.65 ± 0.24

0.11 ± 0.12

0.32 ± 0.03

HV: High Viscosity; O: oxygen; Ca: calcium; C: carbon; P: phosphorus; N: nitrogen; Mg: magnesium; Na: sodium; F: fluorine; Si: silicon; Sr: strontium; S: sulphur. Statistical analysis: ANOVA + Tukey's test (O, P, N, Na, F) and the Kruskal-Wallis + Dwass-Steel-Critchlow-Fligner test (Ca, C, Sr, Mg, S, Si). a: p<0.05 vs. GrandioSO®; b: p<0.05 vs Riva Light Cure; c: p<0.05 vs. Riva Self Cure HV.

Table 6. Microhardness values at the material-sound dentin and carious dentin interface. Microhardness of the materials.

MICRODURE

Materials

Material-sound dentin interface

Material- carious dentin interface

T-test*

Riva Light Cure

62.51 ± 6.23 VHN

c

52.78 ± 9.79 VHN c

13.03 ± 2.98 * VHN c

p<0.001

Riva Self Cure HV

82.50 ± 16.80 VHN

36.33 ± 6.23 VHN

34.56 ± 4.31 VHN

p=0.605

Activa BioActive Restorative™

33.27 ± 8.68 VHN

a, b, c

32.97 ± 3.99 VHN a, b

16.98 ± 4.98 * VHN c

p<0.001

GrandioSO®

111.04 ± 11.13 VHN b, c

56.87 ± 8.95 VHN c

18.62 ± 11.71 * VHN c

p<0.001

  • Discussion

line 432
[7,25,28,32+] - what does 32+ mean as a ref?

It’s a mistake: [7,25,28,32]

Line 456 - a bold claim about bioactive materials, but true. Congratulations! According to the new guidelines of the MDR directive in the EU, the manufacturer of the material must submit irrefutable evidence for all its claims regarding medical materials.

Good luck with your further research!

Round 2

Reviewer 1 Report

Comments and Suggestions for Authors

I have no further suggestions and believe this manuscript is now acceptable for publication.